# Recommender Systems with Generative Retrieval

**Shashank Rajput**[⋆]
University of Wisconsin-Madison

**Nikhil Mehta**[⋆]
Google DeepMind

**Anima Singh**
Google DeepMind

**Raghunandan Keshavan**
Google

**Trung Vu**
Google

**Lukasz Heldt**
Google

**Lichan Hong**
Google DeepMind

**Yi Tay**
Google DeepMind

**Vinh Q. Tran**
Google

**Jonah Samost**
Google

**Maciej Kula**
Google DeepMind

**Ed H. Chi**
Google DeepMind

**Maheswaran Sathiamoorthy**
Google DeepMind

## Abstract

Modern recommender systems perform large-scale retrieval by embedding queries and item candidates in the same unified space, followed by approximate nearest neighbor search to select top candidates given a query embedding. In this paper, we propose a novel generative retrieval approach, where the retrieval model autoregressively decodes the identifiers of the target candidates. To that end, we create semantically meaningful tuple of codewords to serve as a Semantic ID for each item. Given Semantic IDs for items in a user session, a Transformer-based sequence-to-sequence model is trained to predict the Semantic ID of the next item that the user will interact with. We show that recommender systems trained with the proposed paradigm significantly outperform the current SOTA models on various datasets. In addition, we show that incorporating Semantic IDs into the sequence-to-sequence model enhances its ability to generalize, as evidenced by the improved retrieval performance observed for items with no prior interaction history.

## 1 Introduction

Recommender systems help users discover content of interest and are ubiquitous in various recommendation domains such as videos [4, 43, 9], apps [3], products [6, 8], and music [18, 19]. Modern recommender systems adopt a retrieve-and-rank strategy, where a set of viable candidates are selected in the retrieval stage, which are then ranked using a ranker model. Since the ranker model works only on the candidates it receives, it is desired that the retrieval stage emits highly relevant candidates.

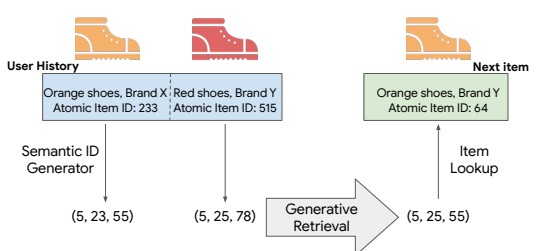

Figure 1: Overview of the *Transformer Index for GEnerative Recommenders* (TIGER) framework. With TIGER, sequential recommendation is expressed as a generative retrieval task by representing each item as a tuple of discrete semantic tokens.

---

[⋆] Equal contribution. Work done when SR was at Google.
Correspondence to rajput.shashank11@gmail.com, nikhilmehta@google.com, nlogn@google.com.

37th Conference on Neural Information Processing Systems (NeurIPS 2023).

There are standard and well-established methods for building retrieval models. Matrix factorization [19] learns query and candidate embeddings in the same space. In order to better capture the non-linearities in the data, dual-encoder architectures [39] (i.e., one tower for the query and another for the candidate) employing inner-product to embed the query and candidate embeddings in the same space have become popular in recent years. To use these models during inference, an index that stores the embeddings for all items is created using the candidate tower. For a given query, its embedding is obtained using the query tower, and an Approximate Nearest Neighbors (ANN) algorithm is used for retrieval. In recent years, the dual encoders architectures have also been extended for sequential recommendations [11, 24, 41, 17, 32, 6, 44] that explicitly take into account the order of user-item interactions.

We propose a new paradigm of building generative retrieval models for sequential recommendation. Instead of traditional query-candidate matching approaches, our method uses an end-to-end generative model that predicts the candidate IDs directly. We propose to leverage the Transformer [36] memory (parameters) as an end-to-end index for retrieval in recommendation systems, reminiscent of Tay et al. [34] that used Transformer memory for document retrieval. We refer to our method as *Transformer Index for GEnerative Recommenders* (TIGER). A high-level overview of TIGER is shown in Figure 1. TIGER is uniquely characterized by a novel semantic representation of items called "Semantic ID" – a sequence of tokens derived from each item's content information. Concretely, given an item's text features, we use a pre-trained text encoder (e.g., SentenceT5 [27]) to generate dense content embeddings. A quantization scheme is then applied on the embedding of an item to form a set of ordered tokens/codewords, which we refer to as the Semantic ID of the item. Ultimately, these Semantic IDs are used to train the Transformer model on the sequential recommendation task.

Representing items as a sequence of semantic tokens has many advantages. Training the transformer memory on semantically meaningful data allows knowledge sharing across similar items. This allows us to dispense away with the atomic and random item Ids that have been previously used [33, 42, 11, 8] as item features in recommendation models. With semantic token representations for items, the model is less prone to the inherent feedback loop [1, 26, 39] in recommendation systems, allowing the model to generalize to newly added items to the corpus. Furthermore, using a sequence of tokens for item representation helps alleviate the challenges associated with the scale of the item corpus; the number of items that can be represented using tokens is the product of the cardinality of each token in the sequence. Typically, the item corpus size can be in the order of billions and learning a unique embedding for each item can be memory-intensive. While random hashing-based techniques [16] can be adopted to reduce the item representation space, in this work, we show that using semantically meaningful tokens for item representation is an appealing alternative. The main contributions of this work are summarized below:

1. We propose TIGER, a novel generative retrieval-based recommendation framework that assigns Semantic IDs to each item, and trains a retrieval model to predict the Semantic ID of an item that a given user may engage with.

2. We show that TIGER outperforms existing SOTA recommender systems on multiple datasets as measured by recall and NDCG metrics.

3. We find that this new paradigm of generative retrieval leads to two additional capabilities in sequential recommender systems: 1. Ability to recommend new and infrequent items, thus improving cold-start recommendations, and 2. Ability to generate diverse recommendations using a tunable parameter.

**Paper Overview.** In Section 2, we provide a brief literature survey of recommender systems, generative retrieval, and the Semantic ID generation techniques we use in this paper. In Section 3, we explain our proposed framework, and outline the various techniques we use for Semantic ID generation. We present the result of our experiments in Section 4, and conclude the paper in Section 5.

## 2 Related Work

**Sequential Recommenders.** Using deep sequential models in recommender systems has developed into a rich literature. GRU4REC [11] was the first to use GRU based RNNs for sequential recommendations. Li et al. [24] proposed Neural Attentive Session-based Recommendation (NARM), where an attention mechanism along with a GRU layer is used to track long term intent of the user.

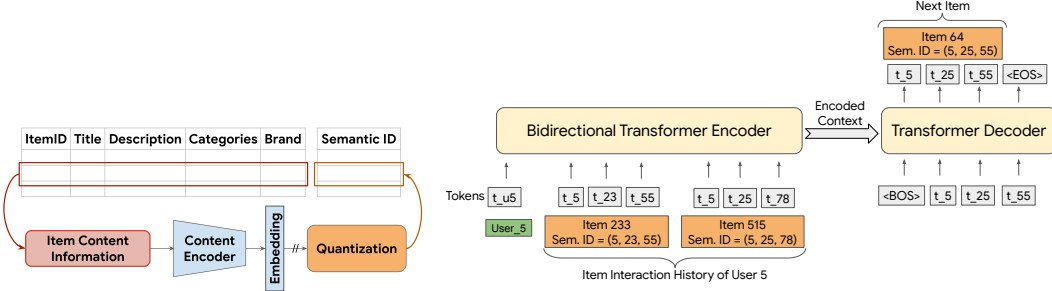

(a) Semantic ID generation for items using quantization of content embeddings.

(b) Transformer based encoder-decoder setup for building the sequence-to-sequence model used for generative retrieval.

Figure 2: An overview of the modeling approach used in TIGER.

AttRec [41] proposed by Zhang et al. used self-attention mechanism to model the user's intent in the current session, and personalization is ensured by modeling user-item affinity with metric learning. Concurrently, Kang et al. also proposed SASRec [17], which used self-attention similar to decoder-only transformer models. Inspired by the success of masked language modeling in language tasks, BERT4Rec [32] and Transformers4Rec [6] utilize transformer models with masking strategies for sequential recommendation tasks. $S^3$-Rec [44] goes beyond just masking by pre-training on four self-supervised tasks to improve data representation. The models described above learn a high-dimensional embedding for each item and perform an ANN in a Maximum Inner Product Search (MIPS) space to predict the next item. In contrast, our proposed technique, TIGER, uses Generative Retrieval to directly predict the Semantic ID of the next item.

P5 [8] fine-tunes a pre-trained large language models for *multi-task* recommender systems. The P5 model relies on the LLM tokenizer (SentencePiece tokenizer [29]) to generate tokens from randomly-assigned item IDs. Whereas, we use Semantic ID representation of items thay are learned based on the content information of the items. In our experiments (Table 2), we demonstrate that recommendation systems based on Semantic ID representation of items yield much better results than using random codes.

**Semantic IDs.** Hou *et al.* proposed VQ-Rec [12] to generate "codes" (analogous to Semantic IDs) using content information for item representation. However, their focus is on building transferable recommender systems, and do not use the codes in a generative manner for retrieval. While they also use product quantization [15] to generate the codes, we use RQ-VAE to generate Semantic IDs, which leads to hierarchical representation of items (Section 4.2). In a concurrent work to us, Singh et al. [31] show that hierarchical Semantic IDs can be used to replace item IDs for ranking models in large scale recommender systems improves model generalization.

**Generative Retrieval.** While techniques for learning search indices have been proposed in the past [20], generative retrieval is a recently developed approach for document retrieval, where the task is to return a set of relevant documents from a database. Some examples include GENRE [5], DSI [34], NCI [37], and CGR [22]. A more detailed coverage of the related work is in Appendix A. To the best of our knowledge, we are the first to propose generative retrieval for recommendation systems using Semantic ID representation of items.

## 3 Proposed Framework

Our proposed framework consists of two stages:

1. *Semantic ID generation using content features.* This involves encoding the item content features to embedding vectors and quantizing the embedding into a tuple of semantic codewords. The resulting tuple of codewords is referred to as the item's Semantic ID.

2. *Training a generative recommender system on Semantic IDs.* A Transformer model is trained on the sequential recommendation task using sequences of Semantic IDs.

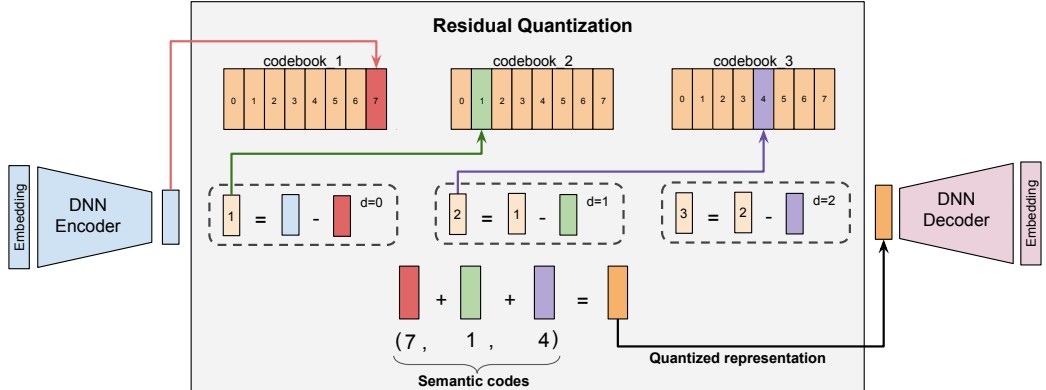

Figure 3: RQ-VAE: In the figure, the vector output by the DNN Encoder, say $r_0$ (represented by the blue bar), is fed to the quantizer, which works iteratively. First, the closest vector to $r_0$ is found in the first level codebook. Let this closest vector be $e_{c_0}$ (represented by the red bar). Then, the residual error is computed as $r_1 := r_0 - e_{c_0}$. This is fed into the second level of the quantizer, and the process is repeated: The closest vector to $r_1$ is found in the second level, say $e_{c_1}$ (represented by the green bar), and then the second level residual error is computed as $r_2 = r_1 - e'_{c_1}$. Then, the process is repeated for a third time on $r_2$. The semantic codes are computed as the indices of $e_{c_0}, e_{c_1}$, and $e_{c_2}$ in their respective codebooks. In the example shown in the figure, this results in the code $(7, 1, 4)$.

## 3.1 Semantic ID Generation

In this section, we describe the Semantic ID generation process for the items in the recommendation corpus. We assume that each item has associated content features that capture useful semantic information (*e.g.* titles or descriptions or images). Moreover, we assume that we have access to a pre-trained content encoder to generate a semantic embedding $x \in \mathbb{R}^d$. For example, general-purpose pre-trained text encoders such as Sentence-T5 [27] and BERT [7] can be used to convert an item's text features to obtain a semantic embedding. The semantic embeddings are then quantized to generate a Semantic ID for each item. Figure 2a gives a high-level overview of the process.

We define a Semantic ID to be a tuple of codewords of length $m$. Each codeword in the tuple comes from a different codebook. The number of items that the Semantic IDs can represent uniquely is thus equal to the product of the codebook sizes. While different techniques to generate Semantic IDs result in the IDs having different semantic properties, we want them to at least have the following property: *Similar items (items with similar content features or whose semantic embeddings are close) should have overlapping Semantic IDs.* For example, an item with Semantic ID $(10, 21, 35)$ should be more similar to one with Semantic ID $(10, 21, 40)$, than an item with ID $(10, 23, 32)$. Next, we discuss the quantization schemes which we use for Semantic ID generation.

**RQ-VAE for Semantic IDs.** Residual-Quantized Variational AutoEncoder (RQ-VAE) [40] is a multi-level vector quantizer that applies quantization on residuals to generate a tuple of codewords (aka Semantic IDs). The Autoencoder is jointly trained by updating the quantization codebook and the DNN encoder-decoder parameters. Fig. 3 illustrates the process of generating Semantic IDs through residual quantization.

RQ-VAE first encodes the input $x$ via an encoder $\mathcal{E}$ to learn a latent representation $z := \mathcal{E}(x)$. At the zero-th level ($d = 0$), the initial residual is simply defined as $r_0 := z$. At each level $d$, we have a codebook $\mathcal{C}_d := \{e_k\}_{k=1}^K$, where $K$ is the codebook size. Then, $r_0$ is quantized by mapping it to the nearest embedding from that level's codebook. The index of the closest embedding $e_{c_d}$ at $d = 0$, i.e., $c_0 = \arg\min_i \|r_0 - e_k\|$, represents the zero-th codeword. For the next level $d = 1$, the residual is defined as $r_1 := r_0 - e_{c_0}$. Then, similar to the zero-th level, the code for the first level is computed by finding the embedding in the codebook for the first level which is nearest to $r_1$. This process is repeated recursively $m$ times to get a tuple of $m$ codewords that represent the Semantic ID. This recursive approach approximates the input from a coarse-to-fine granularity. Note that we chose to use a separate codebook of size $K$ for each of the $m$ levels, instead of using a single, $mK$-sized codebook. This was done because the norm of residuals tends to decrease with increasing levels, hence allowing for different granularities for different levels.

Once we have the Semantic ID $(c_0, \ldots, c_{m-1})$, a quantized representation of $z$ is computed as $\widehat{z} := \sum_{d=0}^{m-1} e_{c_i}$. Then $\widehat{z}$ is passed to the decoder, which tries to recreate the input $x$ using $\widehat{z}$. The RQ-VAE loss is defined as $\mathcal{L}(x) := \mathcal{L}_{\text{recon}} + \mathcal{L}_{\text{rqvae}}$, where $\mathcal{L}_{\text{recon}} := \|x - \widehat{x}\|^2$, and $\mathcal{L}_{\text{rqvae}} := \sum_{d=0}^{m-1} \|\text{sg}[r_i] - e_{c_i}\|^2 + \beta\|r_i - \text{sg}[e_{c_i}]\|^2$. Here $\widehat{x}$ is the output of the decoder, and sg is the stop-gradient operation [35]. This loss jointly trains the encoder, decoder, and the codebook.

As proposed in [40], to prevent RQ-VAE from a codebook collapse, where most of the input gets mapped to only a few codebook vectors, we use k-means clustering-based initialization for the codebook. Specifically, we apply the k-means algorithm on the first training batch and use the centroids as initialization.

**Other alternatives for quantization.** A simple alternative to generating Semantic IDs is to use Locality Sensitive Hashing (LSH). We perform an ablation study in Subsection 4.2 where we find that RQ-VAE indeed works better than LSH. Another option is to use k-means clustering hierarchically [34], but it loses semantic meaning between different clusters [37]. We also tried VQ-VAE, and while it performs similarly to RQ-VAE for generating the candidates during retrieval, it loses the hierarchical nature of the IDs which confers many new capabilities that are discussed in Section 4.3.

**Handling Collisions.** Depending on the distribution of semantic embeddings, the choice of codebook size, and the length of codewords, semantic collisions can occur (*i.e.*, multiple items can map to the same Semantic ID). To remove the collisions, we append an extra token at the end of the ordered semantic codes to make them unique. For example, if two items share the Semantic ID $(12, 24, 52)$, we append additional tokens to differentiate them, representing the two items as $(12, 24, 52, 0)$ and $(12, 24, 52, 1)$. To detect collisions, we maintain a lookup table that maps Semantic IDs to corresponding items. Note that collision detection and fixing is done only once after the RQ-VAE model is trained. Furthermore, since Semantic IDs are integer tuples, the lookup table is efficient in terms of storage in comparison to high dimensional embeddings.

## 3.2 Generative Retrieval with Semantic IDs

We construct item sequences for every user by sorting chronologically the items they have interacted with. Then, given a sequence of the form $(\text{item}_1, \ldots, \text{item}_n)$, the recommender system's task is to predict the next item $\text{item}_{n+1}$. We propose a generative approach that directly predicts the Semantic ID of the next item. Formally, let $(c_{i,0}, \ldots, c_{i,m-1})$ be the $m$-length Semantic ID for $\text{item}_i$. Then, we convert the item sequence to the sequence $(c_{1,0}, \ldots, c_{1,m-1}, c_{2,0}, \ldots, c_{2,m-1}, \ldots, c_{n,0}, \ldots, c_{n,m-1})$. The sequence-to-sequence model is then trained to predict the Semantic ID of $\text{item}_{n+1}$, which is $(c_{n+1,0}, \ldots, c_{n+1,m-1})$. Given the generative nature of our framework, it is possible that a generated Semantic ID from the decoder does not match an item in the recommendation corpus. However, as we show in appendix (Fig. 6) the probability of such an event occurring is low. We further discuss how such events can be handled in appendix E.

## 4 Experiments

**Datasets.** We evaluate the proposed framework on three public real-world benchmarks from the Amazon Product Reviews dataset [10], containing user reviews and item metadata from May 1996 to July 2014. In particular, we use three categories of the Amazon Product Reviews dataset for the sequential recommendation task: "Beauty", "Sports and Outdoors", and "Toys and Games". We discuss the dataset statistics and pre-processing in Appendix C.

**Evaluation Metrics.** We use top-k Recall (Recall@K) and Normalized Discounted Cumulative Gain (NDCG@K) with $K = 5, 10$ to evaluate the recommendation performance.

**RQ-VAE Implementation Details.** As discussed in section 3.1, RQ-VAE is used to quantize the semantic embedding of an item. We use the pre-trained Sentence-T5 [27] model to obtain the semantic embedding of each item in the dataset. In particular, we use item's content features such as title, price, brand, and category to construct a sentence, which is then passed to the pre-trained Sentence-T5 model to obtain the item's semantic embedding of 768 dimension.

The RQ-VAE model consists of three components: a DNN encoder that encodes the input semantic embedding into a latent representation, residual quantizer which outputs a quantized representation, and a DNN decoder that decodes the quantized representation back to the semantic input embedding

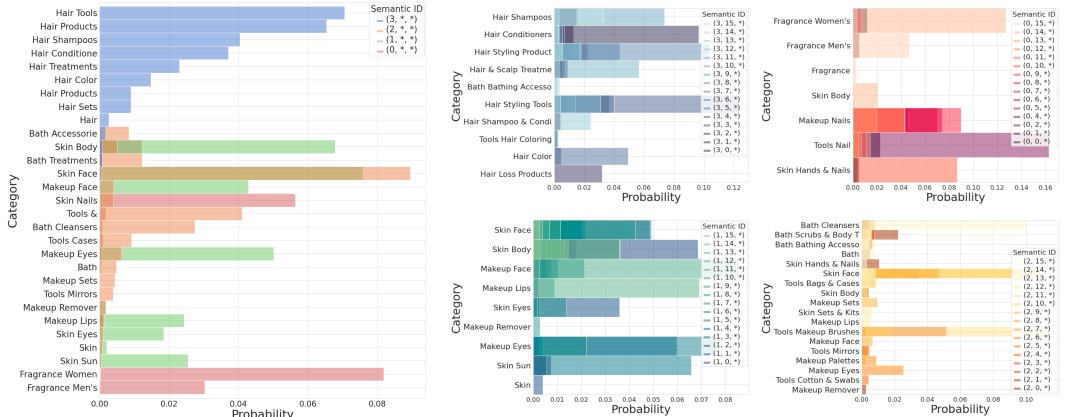

(a) The ground-truth category distribution for all the items in the dataset colored by the value of the first codeword $c_1$.

(b) The category distributions for items having the Semantic ID as $(c_1, *, *)$, where $c_1 \in \{1, 2, 3, 4\}$. The categories are color-coded based on the second semantic token $c_2$.

Figure 4: Qualitative study of RQ-VAE Semantic IDs $(c_1, c_2, c_3, c_4)$ on the Amazon Beauty dataset. We show that the ground-truth categories are distributed across different Semantic tokens. Moreover, the RQVAE semantic IDs form a hierarchy of items, where the first semantic token ($c_1$) corresponds to coarse-level category, while second/third semantic token ($c_2/c_3$) correspond to fine-grained categories.

space. The encoder has three intermediate layers of size 512, 256 and 128 with ReLU activation, with a final latent representation dimension of 32. To quantize this representation, three levels of residual quantization is done. For each level, a codebook of cardinality 256 is maintained, where each vector in the codebook has a dimension of 32. When computing the total loss, we use $\beta = 0.25$. The RQ-VAE model is trained for 20k epochs to ensure high codebook usage ($\geq 80\%$). We use Adagrad optimizer with a learning rate of 0.4 and a batch size of 1024. Upon training, we use the learned encoder and the quantization component to generate a 3-tuple Semantic ID for each item. To avoid multiple items being mapped to the same Semantic ID, we add a unique $4^{th}$ code for items that share the same first three codewords, *i.e.* two items associated with a tuple (7, 1, 4) are assigned (7, 1, 4, 0) and (7, 1, 4, 1) respectively (if there are no collisions, we still assign 0 as the fourth codeword). This results in a unique Semantic ID of length 4 for each item in the recommendation corpus.

**Sequence-to-Sequence Model Implementation Details.** We use the open-sourced T5X framework [28] to implement our transformer based encoder-decoder architecture. To allow the model to process the input for the sequential recommendation task, the vocabulary of the sequence-to-sequence model contains the tokens for each semantic codeword. In particular, the vocabulary contains 1024 ($256 \times 4$) tokens to represent items in the corpus. In addition to the semantic codewords for items, we add user-specific tokens to the vocabulary. To keep the vocabulary size limited, we only add 2000 tokens for user IDs. We use the Hashing Trick [38] to map the raw user ID to one of the 2000 user ID tokens. We construct the input sequence as the user Id token followed by the sequence of Semantic ID tokens corresponding to a given user's item interaction history. We found that adding user ID to the input, allows the model to personalize the items retrieved.

We use 4 layers each for the transformer-based encoder and decoder models with 6 self-attention heads of dimension 64 in each layer. We used the ReLU activation function for all the layers. The MLP and the input dimension was set as 1024 and 128, respectively. We used a dropout of 0.1. Overall, the model has around 13 million parameters. We train this model for 200k steps for the "Beauty" and "Sports and Outdoors" dataset. Due to the smaller size of the "Toys and Games" dataset, it is trained only for 100k steps. We use a batch size of 256. The learning rate is 0.01 for the first 10k steps and then follows an inverse square root decay schedule.

## 4.1 Performance on Sequential Recommendation

In this section, we compare our proposed framework for generative retrieval with the following sequential recommendation methods (which are described briefly in Appendix B): GRU4Rec [11], Caser [33], HGN [25], SASRec [17], BERT4Rec [32], FDSA [42], S$^3$-Rec [44], and P5 [8]. Notably

Table 1: Performance comparison on sequential recommendation. The last row depicts % improvement with TIGER relative to the best baseline. Bold (underline) are used to denote the best (second-best) metric.

| Methods | Sports and Outdoors | | | | Beauty | | | | Toys and Games | | | |
|---|---|---|---|---|---|---|---|---|---|---|---|---|
| | Recall @5 | NDCG @5 | Recall @10 | NDCG @10 | Recall @5 | NDCG @5 | Recall @10 | NDCG @10 | Recall @5 | NDCG @5 | Recall @10 | NDCG @10 |
| P5 [8] | 0.0061 | 0.0041 | 0.0095 | 0.0052 | 0.0163 | 0.0107 | 0.0254 | 0.0136 | 0.0070 | 0.0050 | 0.0121 | 0.0066 |
| Caser [33] | 0.0116 | 0.0072 | 0.0194 | 0.0097 | 0.0205 | 0.0131 | 0.0347 | 0.0176 | 0.0166 | 0.0107 | 0.0270 | 0.0141 |
| HGN [25] | 0.0189 | 0.0120 | 0.0313 | 0.0159 | 0.0325 | 0.0206 | 0.0512 | 0.0266 | 0.0321 | 0.0221 | 0.0497 | 0.0277 |
| GRU4Rec [11] | 0.0129 | 0.0086 | 0.0204 | 0.0110 | 0.0164 | 0.0099 | 0.0283 | 0.0137 | 0.0097 | 0.0059 | 0.0176 | 0.0084 |
| BERT4Rec [32] | 0.0115 | 0.0075 | 0.0191 | 0.0099 | 0.0203 | 0.0124 | 0.0347 | 0.0170 | 0.0116 | 0.0071 | 0.0203 | 0.0099 |
| FDSA [42] | 0.0182 | 0.0122 | 0.0288 | 0.0156 | 0.0267 | 0.0163 | 0.0407 | 0.0208 | 0.0228 | 0.0140 | 0.0381 | 0.0189 |
| SASRec [17] | 0.0233 | 0.0154 | 0.0350 | 0.0192 | 0.0387 | 0.0249 | 0.0605 | 0.0318 | 0.0463 | 0.0306 | 0.0675 | 0.0374 |
| S³-Rec [44] | 0.0251 | 0.0161 | 0.0385 | 0.0204 | 0.0387 | 0.0244 | 0.0647 | 0.0327 | 0.0443 | 0.0294 | 0.0700 | 0.0376 |
| **TIGER [Ours]** | **0.0264** | **0.0181** | **0.0400** | **0.0225** | **0.0454** | **0.0321** | **0.0648** | **0.0384** | **0.0521** | **0.0371** | **0.0712** | **0.0432** |
| | +5.22% | +12.55% | +3.90% | +10.29% | +17.31% | +29.04% | +0.15% | +17.43% | +12.53% | +21.24% | +1.71% | +14.97% |

all the baselines (except P5), learn a high-dimensional vector space using dual encoder, where the user's past item interactions and the candidate items are encoded as a high-dimensional representation and Maximum Inner Product Search (MIPS) is used to retrieve the next candidate item that the user will potentially interact with. In contrast, our *novel* generative retrieval framework directly predicts the item's Semantic ID token-by-token using a sequence-to-sequence model.

**Recommendation Performance.** We perform an extensive analysis of our proposed TIGER on the sequential recommendation task and compare against the baselines above. The results for all baselines, except P5, are taken from the publicly accessible results[3] made available by Zhou *et al.* [44]. For P5, we use the source code made available by the authors. However, for a fair comparison, we updated the data pre-processing method to be consistent with the other baselines and our method. We provide further details related to our changes in Appendix D.

The results are shown in Table 1. We observe that TIGER consistently outperforms the existing baselines[4]. We see significant improvement across all the three benchmarks that we considered. In particular, TIGER performs considerably better on the Beauty benchmark compared to the second-best baseline with up to 29% improvement in NDCG@5 compared to SASRec and 17.3% improvement in Recall@5 compared to S³-Rec. Similarly on the Toys and Games dataset, TIGER is 21% and 15% better in NDCG@5 and NDCG@10, respectively.

## 4.2 Item Representation

In this section, we analyze several important characteristics of RQ-VAE Semantic IDs. In particular, we first perform a qualitative analysis to observe the hierarchical nature of Semantic IDs. Next, we evaluate the importance of our design choice of using RQ-VAE for quantization by contrasting the performance with an alternative hashing-based quantization method. Finally, we perform an ablation to study the importance of using Semantic IDs by comparing TIGER with a sequence-to-sequence model that uses Random ID for item representation.

**Qualitative Analysis.** We analyze the RQ-VAE Semantic IDs learned for the Amazon Beauty dataset in Figure 4. For exposition, we set the number of RQ-VAE levels as 3 with a codebook size of 4, 16, and 256 respectively, *i.e.* for a given Semantic ID $(c_1, c_2, c_3)$ of an item, $0 \le c_1 \le 3$, $0 \le c_2 \le 15$ and $0 \le c_3 \le 255$. In Figure 4a, we annotate each item's category using $c_1$ to visualize $c_1$-specific categories in the overall category distribution of the dataset. As shown in Figure 4a, $c_1$ captures the high-level category of the item. For instance, $c_1 = 3$ contains most of the products related to "Hair". Similarly, majority of items with $c_1 = 1$ are "Makeup" and "Skin" products for face, lips and eyes.

We also visualize the hierarchical nature of RQ-VAE Semantic IDs by fixing $c_1$ and visualizing the category distribution for all possible values of $c_2$ in Fig. 4b. We again found that the second codeword $c_2$ further categorizes the high-level semantics captured with $c_1$ into fine-grained categories. The hierarchical nature of Semantic IDs learned by RQ-VAE opens a wide-array of new capabilities which are discussed in Section 4.3. As opposed to existing recommendation systems that learn item embeddings based on random atomic IDs, TIGER uses Semantic IDs where semantically similar

---

[3]https://github.com/aHuiWang/CIKM2020-S3Rec
[4]We show in Table 9 that the standard error in the metrics for TIGER is insignificant.

Table 2: Ablation study for different ID generation techniques for generative retrieval. We show that RQ-VAE Semantic ID (SID) perform significantly better compared to hashing SIDs and Random IDs.

| Methods | Sports and Outdoors | | | | Beauty | | | | Toys and Games | | | |
|---|---|---|---|---|---|---|---|---|---|---|---|---|
| | Recall @5 | NDCG @5 | Recall @10 | NDCG @10 | Recall @5 | NDCG @5 | Recall @10 | NDCG @10 | Recall @5 | NDCG @5 | Recall @10 | NDCG @10 |
| Random ID | 0.007 | 0.005 | 0.0116 | 0.0063 | 0.0296 | 0.0205 | 0.0434 | 0.0250 | 0.0362 | 0.0270 | 0.0448 | 0.0298 |
| LSH SID | 0.0215 | 0.0146 | 0.0321 | 0.0180 | 0.0379 | 0.0259 | 0.0533 | 0.0309 | 0.0412 | 0.0299 | 0.0566 | 0.0349 |
| RQ-VAE SID | **0.0264** | **0.0181** | **0.0400** | **0.0225** | **0.0454** | **0.0321** | **0.0648** | **0.0384** | **0.0521** | **0.0371** | **0.0712** | **0.0432** |

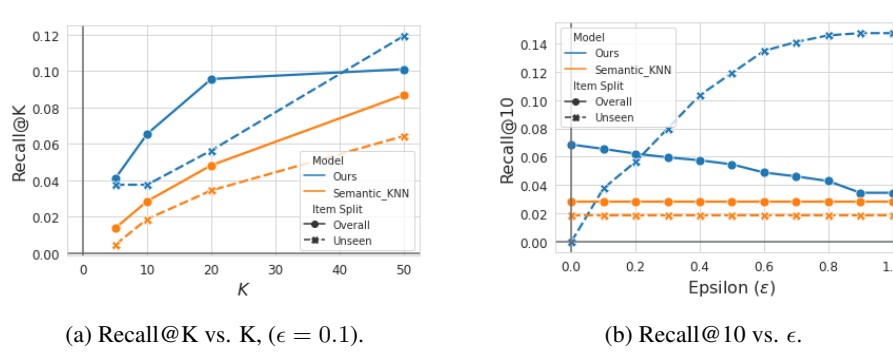

(a) Recall@K vs. K, ($\epsilon = 0.1$).      (b) Recall@10 vs. $\epsilon$.

Figure 5: Performance in the cold-start retrieval setting.

items have overlapping codewords, which allows the model to effectively share knowledge from semantically similar items in the dataset.

**Hashing vs. RQ-VAE Semantic IDs.** We study the importance of RQ-VAE in our framework by comparing RQ-VAE against Locality Sensitive Hashing (LSH) [14, 13, 2] for Semantic ID generation. LSH is a popular hashing technique that can be easily adapted to work for our setting. To generate LSH Semantic IDs, we use $h$ random hyperplanes $\boldsymbol{w}_1, \ldots, \boldsymbol{w}_h$ to perform a random projection of the embedding vector $\boldsymbol{x}$ and compute the following binary vector: $(1_{\boldsymbol{w}_1^\top \boldsymbol{x} > 0}, \ldots, 1_{\boldsymbol{w}_h^\top \boldsymbol{x} > 0})$. This vector is converted into an integer code as $c_0 = \sum_{i=1}^{h} 2^{i-1} 1_{\boldsymbol{w}_i^\top \boldsymbol{x} > 0}$. This process is repeated $m$ times using an independent set of random hyperplanes, resulting in $m$ codewords $(c_0, c_1, \ldots, c_{m-1})$, which we refer to as the LSH Semantic ID.

In Table 2, we compare the performance of LSH Semantic ID with our proposed RQ-VAE Semantic ID. In this experiment, for LSH Semantic IDs, we used $h = 8$ random hyperplanes and set $m = 4$ to ensure comparable cardinality with the RQ-VAE. The parameters for the hyperplanes are randomly sampled from a standard normal distribution, which ensures that the hyperplanes are spherically symmetric. Our results show that RQ-VAE consistently outperforms LSH. This illustrates that learning Semantic IDs via a non-linear, Deep Neural Network (DNN) architecture yields better quantization than using random projections, given the same content-based semantic embedding.

**Random ID vs. Semantic ID.** We also compare the importance of Semantic IDs in our generative retrieval recommender system. In particular, we compare randomly generated IDs with the Semantic IDs. To generate the Random ID baseline, we assign $m$ random codewords to each item. A Random ID of length $m$ for an item is simply $(c_1, \ldots, c_m)$, where $c_i$ is sampled uniformly at random from $\{1, 2, \ldots, K\}$. We set $m = 4$, and $K = 255$ for the Random ID baseline to make the cardinality similar to RQ-VAE Semantic IDs. A comparison of Random ID against RQ-VAE and LSH Semantic IDs is shown in Table 2. We see that Semantic IDs consistently outperform Random ID baseline, highlighting the importance of leveraging content-based semantic information.

### 4.3 New Capabilities

We describe two new capabilities that directly follow from our proposed generative retrieval framework, namely cold-start recommendations and recommendation diversity. We refer to these capabilities as "new" since existing sequential recommendation models (See the baselines in section 4.1) cannot be directly used to satisfy these real-world use cases. These capabilities result from a synergy between RQ-VAE based Semantic IDs and the generative retrieval approach of our framework. We discuss how TIGER is used in these settings in the following sections.

**Cold-Start Recommendation.** In this section, we study the cold-start recommendation capability of our proposed framework. Due to the fast-changing nature of the real-world recommendation corpus, new items are constantly introduced. Since newly added items lack user impressions in the training corpus, existing recommendation models that use a random atomic ID for item representation fail to retrieve new items as potential candidates. In contrast, the TIGER framework can easily perform cold-start recommendations since it leverages item semantics when predicting the next item.

For this analysis, we consider the Beauty dataset from Amazon Reviews. To simulate newly added items, we remove 5% of test items from the training data split. We refer to these removed items as *unseen items*. Removing the items from the training split ensures there is no data leakage with respect to the unseen items. As before, we use Semantic ID of length 4 to represent the items, where the first 3 tokens are generated using RQ-VAE and the $4^{th}$ token is used to ensure a unique ID exists for all the seen items. We train the RQ-VAE quantizer and the sequence-to-sequence model on the training split. Once trained, we use the RQ-VAE model to generate the Semantic IDs for all the items in the dataset, including any unseen items in the item corpus.

Given a Semantic ID $(c_1, c_2, c_3, c_4)$ predicted by the model, we retrieve the seen item having the same corresponding ID. Note that by definition, each Semantic ID predicted by the model can match at most one item in the training dataset. Additionally, unseen items having the same first three semantic tokens, *i.e.* $(c_1, c_2, c_3)$ are included to the list of retrieved candidates. Finally, when retrieving a set of top-K candidates, we introduce a hyperparameter $\epsilon$ which specifies the maximum proportion of unseen items chosen by our framework.

We compare the performance of TIGER with the k-Nearest Neighbors (KNN) approach on the cold-start recommendation setting in Fig. 5. For KNN, we use the semantic representation space to perform the nearest-neighbor search. We refer to the KNN-based baseline as Semantic_KNN. Fig. 5a shows that our framework with $\epsilon = 0.1$ consistently outperforms Semantic_KNN for all Recall@K metrics. In Fig. 5b, we provide a comparison between our method and Semantic_KNN for various values of $\epsilon$. For all settings of $\epsilon \geq 0.1$, our method outperforms the baseline.

**Recommendation diversity.** While Recall and NDCG are the primary metrics used to evaluate a recommendation system, diversity of predictions is another critical objective of interest. A recommender system with poor diversity can be detrimental to the long-term engagement of users. Here, we discuss how our generative retrieval framework can be used

Table 3: The entropy of the category distribution predicted by the model for the Beauty dataset. A higher entropy corresponds more diverse items predicted by the model.

| Temperature | Entropy@10 | Entropy@20 | Entropy@50 |
|---|---|---|---|
| T = 1.0 | 0.76 | 1.14 | 1.70 |
| T = 1.5 | 1.14 | 1.52 | 2.06 |
| T = 2.0 | 1.38 | 1.76 | 2.28 |

to predict diverse items. We show that temperature-based sampling during the decoding process can be effectively used to control the diversity of model predictions. While temperature-based sampling can be applied to any existing recommendation model, TIGER allows sampling across various levels of hierarchy owing to the properties of RQ-VAE Semantic IDs. For instance, sampling the first token of the Semantic ID allows retrieving items from coarse-level categories, while sampling a token from second/third token allows sampling items within a category.

Table 4: Recommendation diversity with temperature-based decoding.

| Target Category | Most-common Categories for top-10 predicted items | |
|---|---|---|
| | T = 1.0 | T = 2.0 |
| Hair Styling Products | Hair Styling Products | Hair Styling Products, Hair Styling Tools, Skin Face |
| Tools Nail | Tools Nail | Tools Nail, Makeup Nails |
| Makeup Nails | Makeup Nails | Makeup Nails, Skin Hands & Nails, Tools Nail |
| Skin Eyes | Skin Eyes | Hair Relaxers, Skin Face, Hair Styling Products, Skin Eyes |
| Makeup Face | Tools Makeup Brushes,Makeup Face | Tools Makeup Brushes, Makeup Face,Skin Face, Makeup Sets, Hair Styling Tools |
| Hair Loss Products | Hair Loss Products,Skin Face, Skin Body | Skin Face, Hair Loss Products, Hair Shampoos,Hair & Scalp Treatments, Hair Conditioners |

We quantitatively measure the diversity of predictions using Entropy@K metric, where the entropy is calculated for the distribution of the ground-truth categories of the top-K items predicted by the model. We report the Entropy@K for various temperature values in Table 3. We observe that temperature-sampling in the decoding stage can be effectively used to increase the diversity in the ground-truth categories of the items. We also perform a qualitative analysis in Table 4.

Table 5: Recall and NDCG metrics for different number layers.

| Number of Layers | Recall@5 | NDCG@5 | Recall@10 | NDCG@10 |
|---|---|---|---|---|
| 3 | 0.04499 | 0.03062 | 0.06699 | 0.03768 |
| 4 | 0.0454 | 0.0321 | 0.0648 | 0.0384 |
| 5 | 0.04633 | 0.03206 | 0.06596 | 0.03834 |

### 4.4 Ablation Study

We measure the effect of varying the number of layers in the sequence-to-sequence model in Table 5. We see that the metrics improve slightly as we make the network bigger. We also measure the effect of providing user information, the results for which are provided in Table 8 in the Appendix.

### 4.5 Invalid IDs

Since the model decodes the codewords of the target Semantic ID autoregressively, it is possible that the model may predict invalid IDs (i.e., IDs that do not map to any item in the recommendation dataset). In our experiments, we used semantic IDs of length $4$ with each codeword having a cardinality of $256$ (i.e., codebook size = 256 for each level). The number of possible IDs spanned by this combination $= 256^4$, which is approximately 4 trillion. On the other hand, the number of items in the datasets we consider is 10K-20K (See Table 6). Even though the number of valid IDs is only a fraction of all complete ID space, we observe that the model almost always predicts valid IDs. We visualize the fraction of invalid IDs produced by TIGER as a function of the number of retrieved items $K$ in Figure 6. For top-10 predictions, the fraction of invalid IDs varies from $\sim 0.1\% - 1.6\%$ for the three datasets. To counter the effect of invalid IDs and to always get top-10 valid IDs, we can increase the beam size and filter the invalid IDs.

It is important to note that, despite generating invalid IDs, TIGER achieves state-of-the-art performance when compared to other popular methods used for sequential recommendations. One extension to handle invalid tokens could be to do prefix matching when invalid tokens are generated by the model. Prefix matching of Semantic IDs would allow retrieving items that have similar semantic meaning as the tokens generated by the model. Given the hierarchical nature of our RQ-VAE tokens, prefix matching can be thought of as model predicting item category as opposed to the item index. Note that such an extension could improve the recall/NDCG metrics even further. We leave such an extension as a future work.

## 5 Conclusion

This paper proposes a novel paradigm, called TIGER, to retrieve candidates in recommender systems using a generative model. Underpinning this method is a novel semantic ID representation for items, which uses a hierarchical quantizer (RQ-VAE) on content embeddings to generate tokens that form the semantic IDs. Our framework provides results in a model that can be used to train and serve without creating an index — the transformer memory acts as a semantic index for items. We note that the cardinality of our embedding table does not grow linearly with the cardinality of item space, which works in our favor compared to systems that need to create large embedding tables during training or generate an index for every single item. Through experiments on three datasets, we show that our model can achieve SOTA retrieval performance, while generalizing to new and unseen items.

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

# A Related Work (cont.)

**Generative Retrieval** Document retrieval traditionally involved training a 2-tower model which mapped both queries and documents to the same high-dimensional vector space, followed by performing an ANN or MIPS for the query over all the documents to return the closest ones. This technique presents some disadvantages like having a large embedding table [22, 23]. Generative retrieval is a recently proposed technique that aims to fix some of the issues of the traditional approach by producing token by token either the title, name, or the document id string of the document. Cao et al. [5] proposed GENRE for entity retrieval, which used a transformer-based architecture to return, token-by-token, the name of the entity referenced to in a given query. Tay et al. [34] proposed DSI for document retrieval, which was the first system to assign structured semantic DocIDs to each document. Then given a query, the model autoregressively returned the DocID of the document token-by-token. The DSI work marks a paradigm shift in IR to generative retrieval approaches and is the first successful application of an end-to-end Transformer for retrieval applications. Subsequently, Lee et al. [23] show that generative document retrieval is useful even in the multi-hop setting, where a complex query cannot be answered directly by a single document, and hence their model generates intermediate queries, in a chain-of-thought manner, to ultimately generate the output for the complex query. Wang et al. [37] supplement the hierarchical $k$-means clustering based semantic DocIDs of Tay et al. [34] by proposing a new decoder architecture that specifically takes into account the prefixes in semantic DocIDs. In CGR [22], the authors propose a way to take advantage of both the bi-encoder technique and the generative retrieval technique, by allowing the decoder, of their encoder-decoder-based model, to learn separate *contextualized* embeddings which store information about documents intrinsically. To the best of our knowledge, we are the first to use generative Semantic IDs created using an auto-encoder (RQ-VAE [40, 21]) for retrieval models.

**Vector Quantization.** We refer to Vector Quantization as the process of converting a high-dimensional vector into a low-dimensional tuple of codewords. One of the most straightforward techniques uses hierarchical clustering, such as the one used in [34], where clusters created in a particular iteration are further partitioned into sub-clusters in the next iteration. An alternative popular approach is Vector-Quantized Variational AutoEncoder (VQ-VAE), which was introduced in [35] as a way to encode natural images into a sequence of codes. The technique works by first passing the input vector (or image) through an encoder that reduces the dimensionality. The smaller dimensional vector is partitioned and each partition is quantized separately, thus resulting in a sequence of codes: one code per partition. These codes are then used by a decoder to recreate the original vector (or image).

RQ-VAE [40, 21] applies residual quantization to the output of the encoder of VQ-VAE to achieve a lower reconstruction error. We discuss this technique in more detail in Subsection 3.1. Locality Sensitive Hashing (LSH) [14, 13] is a popular technique used for clustering and approximate nearest neighbor search. The particular version that we use in this paper for clustering is SimHash [2], which uses random hyperplanes to create binary vectors which serve as hashes of the items. Because it has low computational complexity and is scalable [13], we use this as a baseline technique for vector quantization.

# B Baselines

Below we briefly describe each of the baselines with which we compare TIGER

- GRU4Rec [11] is the first RNN-based approach that uses a customized GRU for the sequential recommendation task.
- Caser [33] uses a CNN architecture for capturing high-order Markov Chains by applying horizontal and vertical convolutional operations for sequential recommendation.
- HGN [25]: Hierarchical Gating Network captures the long-term and short-term user interests via a new gating architecture.
- SASRec [17]: Self-Attentive Sequential Recommendation uses a causal mask Transformer to model a user's sequential interactions.
- BERT4Rec [32]: BERT4Rec addresses the limitations of uni-directional architectures by using a bi-directional self-attention Transformer for the recommendation task.

- FDSA [42]: Feature-level Deeper Self-Attention Network incorporates item features in addition to the item embeddings as part of the input sequence in the Transformers.

- S$^3$-Rec [44]: Self-Supervised Learning for Sequential Recommendation proposes pre-training a bi-directional Transformer on self-supervision tasks to improve the sequential recommendation.

- P5 [8]: P5 is a recent method that uses a pretrained Large Language Model (LLM) to unify different recommendation tasks in a single model.

## C   Dataset Statistics

Table 6: Dataset statistics for the three real-world benchmarks.

| Dataset | # Users | # Items | Sequence Length | |
|---|---|---|---|---|
| | | | Mean | Median |
| Beauty | 22,363 | 12,101 | 8.87 | 6 |
| Sports and Outdoors | 35,598 | 18,357 | 8.32 | 6 |
| Toys and Games | 19,412 | 11,924 | 8.63 | 6 |

We use three public benchmarks from the Amazon Product Reviews dataset [10], containing user reviews and item metadata from May 1996 to July 2014. We use three categories of the Amazon Product Reviews dataset for the sequential recommendation task: "Beauty", "Sports and Outdoors", and "Toys and Games". Table 6 summarizes the statistics of the datasets. We use users' review history to create item sequences sorted by timestamp and filter out users with less than 5 reviews. Following the standard evaluation protocol [17, 8], we use the leave-one-out strategy for evaluation. For each item sequence, the last item is used for testing, the item before the last is used for validation, and the rest is used for training. During training, we limit the number of items in a user's history to 20.

## D   Modifications to the P5 data preprocessing

Table 7: Results for P5[8] with the standard pre-processing.

| Methods | Sports and Outdoors | | | | Beauty | | | | Toys and Games | | | |
|---|---|---|---|---|---|---|---|---|---|---|---|---|
| | Recall@5 | NDCG@5 | Recall@10 | NDCG@10 | Recall@5 | NDCG@5 | Recall@10 | NDCG@10 | Recall@5 | NDCG@5 | Recall@10 | NDCG@10 |
| P5 | 0.0061 | 0.0041 | 0.0095 | 0.0052 | 0.0163 | 0.0107 | 0.0254 | 0.0136 | 0.0070 | 0.0050 | 0.0121 | 0.0066 |
| P5-ours | 0.0107 | 0.0076 | 0.01458 | 0.0088 | 0.035 | 0.025 | 0.048 | 0.0298 | 0.018 | 0.013 | 0.0235 | 0.015 |

The P5 source code [5] pre-processes the Amazon dataset to first create sessions for each user, containing chronologically ordered list of items that the user reviewed. After creating these sessions, the original item IDs from the dataset are remapped to integers $1, 2, 3, \ldots$ [6]. Hence, the first item in the first session gets an id of '1', the second item, if not seen before, gets an id of '2', and so on. Notably, this pre-processing scheme is applied before creating training and testing splits. This results in the creation of a sequential dataset where many sequences are of the form $a, a + 1, a + 2, \ldots$. Given that P5 uses Sentence Piece tokenizer [30] (See Section 4.1 in [8]), the test and train items in a user session may share the sub-word and can lead to information leakage during inference.

To resolve the leakage issue, instead of assigning sequentially increasing integer ids to items, we assigned random integer IDs, and then created splits for training and evaluation. The rest of the code for P5 was kept identical to the source code provided in the paper. The results for this dataset are reported in Table 7 as the row 'P5'. We also implemented a version of P5 ourselves from scratch, and train the model on only sequential recommendation task. The results for our implementation are depicted as 'P5-ours'. We were also able to verify in our P5 implementation that using consecutive integer sequences for the item IDs helped us get equivalent or better metrics than those reported in P5.

Table 8: The effect of providing user information to the recommender system

| Recall@5 | NDCG@5 | Recall@10 | NDCG@10 |
|---|---|---|---|
| No user information | 0.04458 | 0.0302 | 0.06479 | 0.0367 |
| With user id (reported in the paper) | 0.0454 | 0.0321 | 0.0648 | 0.0384 |

Table 9: The mean and stand error of the metrics for different dataset (computed using 3 runs with different random seeds)

| Datasets | Recall@5 | NDCG@5 | Recall@10 | NDCG@10 |
|---|---|---|---|---|
| Beauty | $0.0441 \pm 0.00069$ | $0.0309 \pm 0.00062$ | $0.0642 \pm 0.00092$ | $0.0374 \pm 0.00061$ |
| Sports and Outdoors | $0.0278 \pm 0.00069$ | $0.0189 \pm 0.00043$ | $0.0419 \pm 0.0010$ | $0.0234 \pm 0.00048$ |
| Toys and Games | $0.0518 \pm 0.00064$ | $0.0375 \pm 0.00039$ | $0.0698 \pm 0.0013$ | $0.0433 \pm 0.00047$ |

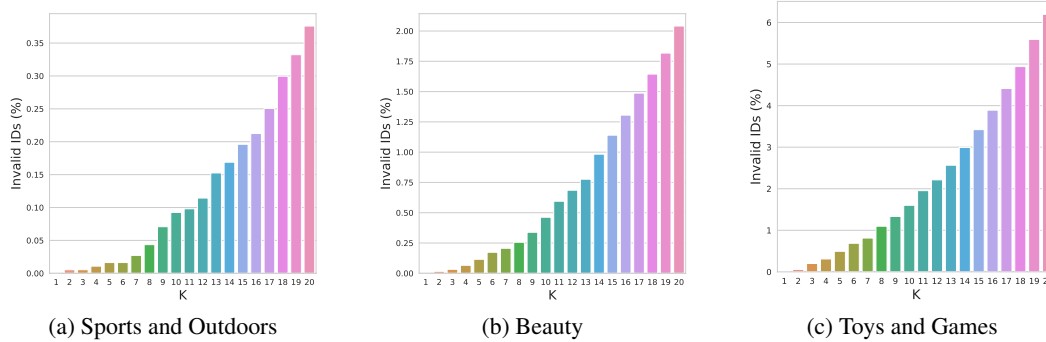

(a) Sports and Outdoors      (b) Beauty      (c) Toys and Games

Figure 6: Percentage of invalid IDs when generating Semantic IDs using Beam search for various values of $K$. As shown, $\sim 0.3\% - 6\%$ of the IDs are invalid when retrieving the top-20 items.

## E  Discussion

**Effects of Semantic ID length and codebook size.** We tried varying the Semantic ID length and codebook size, such as having an ID consisting of 6 codewords each from a codebook of size 64. We noticed that the recommendation metrics for TIGER were robust to these changes. However, note that the input sequence length increases with longer IDs (i.e., more codewords per item ID), which makes the computation more expensive for our transformer-based sequence-to-sequence model.

**Scalability.** To test the scalability of Semantic IDs, we ran the following experiment: We combined all the three datasets and generated Semantic IDs for the entire set of items from the three datasets. Then, we used these Semantic IDs for the recommendation task on the Beauty dataset. We compare the results from this experiment with the original experiment where the Semantic IDs are generated only from the Beauty dataset. The results are provided in Table 10. We see that there is only a small decrease in performance here.

**Inference cost.** Despite the remarkable success of our model on the sequential recommendation task, we note that our model can be more computationally expensive than ANN-based models during inference due to the use of beam search for autoregressive decoding. We emphasize that optimizing the computational efficiency of TIGER was not the main objective of this work. Instead, our work opens up a new area of research: *Recommender Systems based on Generative Retrieval*. As part of future work, we will consider ways to make the model smaller or explore other ways of improving the inference efficiency.

**Memory cost of lookup tables.** We maintain two lookup hash tables for TIGER: an Item ID to Semantic ID table and a Semantic ID to Item ID table. Note that both of these tables are generated only once and then frozen: they are generated after the training of the RQ-VAE-based Semantic ID generation model, and after that, they are frozen for the training of the sequence-to-sequence

---

[5] https://github.com/jeykigung/P5

[6] https://github.com/jeykigung/P5/blob/0aaa3fd8366bb6e708c8b70708291f2b0ae90c82/preprocess/data_preprocess_amazon.ipynb

Table 10: Testing scalability by generating the Semantic IDs on the combined dataset vs generating the Semantic IDs on only the Beauty dataset.

|  | Recall@5 | NDCG@5 | Recall@10 | NDCG@10 |
|---|---|---|---|---|
| Semantic ID [Combined datasets] | 0.04355 | 0.3047 | 0.06314 | 0.03676 |
| Semantic ID [Amazon Beauty] | 0.0454 | 0.0321 | 0.0648 | 0.0384 |

transformer model. Each Semantic ID consists of a tuple of 4 integers, each of which are stored in 8 bits, hence totalling to 32 bits per item. Each item is represented by an Item ID, stored as a 32 bit integer. Thus, the size of each lookup table will be of the order of $64N$ bits, where $N$ is the number of items in the dataset.

**Memory cost of embedding tables**. TIGER uses much smaller embedding tables compared to traditional recommender systems. This is because where traditional recommender systems store an embedding for each item, TIGER only stores an embedding for each semantic codeword. In our experiments, we used 4 codewords each of cardinality 256 for Semantic ID representation, resulting in 1024 ($256{\times}4$) embeddings. For traditional recommender systems, the number of embeddings is $N$, where $N$ is the number of items in the dataset. In our experiments, $N$ ranged from 10K to 20K depending on the dataset. Hence, the memory cost of TIGER's embedding table is $1024d$, where $d$ is the dimension of the embedding, whereas the memory cost for embedding lookup tables in traditional recommendation systems is $Nd$.

