# OpenReview forum: "Recommender Systems with Generative Retrieval"
_NeurIPS.cc/2023/Conference — NeurIPS 2023 poster_

### Official Review · Reviewer_5EdZ · 2023-06-28

**Soundness:** 3 good
**Presentation:** 3 good
**Contribution:** 1 poor
**Rating:** 3
**Confidence:** 4

**Summary:**

This paper proposes a generative retrieval approach wherein the model autoregressively decodes the identifiers, called Semantic ID, for the target candidate. They propose a method to generate the semantic ID for items which is a tuple of codewords which are trained from the embeddings of the text component of the items obtained using a standard sentence BERT type of model. Finally, a seq2seq model is trained to predict the semantic ids for the next item given the current item. The authors show that the proposed technique outperforms SOTA techniques on some retrieval datasets. They also claim that the semantic ID trick enhances the generalization property of the seq2seq model especially for cold start cases.

**Strengths:**

Originality: The paper seems to present a novel idea which is original enough. The items are represented in a novel semantic ID sequence form which is then used in a standard seq2seq framework.

Clarity: I think the paper is clear enough to understand and reproduce.



**Weaknesses:**

The method itself is and its larger claim of generative in nature is the primary concern I have with the paper. The proposed method tries to generate semantic id sequence for items already seen by the system. Any novel combination of semantic ID will lead to an item which is not part of the retrieval system. This leads to the authors constraining the semantic IDs to just 3 code words and then an ad hoc 4th code word to introduce uniqueness in the codes. This part seems to suit small, well controlled item sets, but the scaling out of such a system to billions of items seems not clearly explained or experimented with.

The experimental results seem not consistent with the published paper, especially for P5 paper. The numbers reported in the paper are higher than this paper which are not shown as is, but the authors implement their own P5 and compare results which are way different than the published results.

The other comparative results are all copied over from the benchmark mentioned in the footnote ttps://github.com/aHuiWang/CIKM2020-S3Rec.

**Questions:**

I would like to see large code sets for items. This would make the system truly scalable. Otherwise, constraining embeddings to 3 codewords seems a step in the opposite direction which will work with very small item sets in controlled environments.

I would also ask the authors to perform comparative experiments with P5, where the parameters do not need to be comparable, but the best performance obtained should be the primary aim.

---

> ### Author Rebuttal · Authors · 2023-08-10
>
> We thank the reviewer for their time and effort in reviewing our paper and providing valuable feedback. We are glad to know that the reviewer found our idea novel, and the paper easy to understand and reproduce. Please find our responses to your questions below:
>
> > I would like to see large code sets for items. This would make the system truly scalable. Otherwise, constraining embeddings to 3 codewords seems a step in the opposite direction which will work with very small item sets in controlled environments.
>
> Thank you for the suggestion. Scaling with Semantic IDs can be easily achieved by increasing the size/cardinality of the codebook used in the RQ-VAE model. In our experiments, we used 256 as the codebook size. Note that theoretically the number of items that can be uniquely represented with Semantic IDs grow exponentially with codebook cardinality. For instance, three tokens with 256 cardinality can potentially represent $256^3$ items. However, since Semantic IDs are generated using content features, multiple items may have the same Semantic ID (referred to as a collision) when the items have similar content features. This can be easily mitigated by adding an additional token that uniquely identifies items that undergo collisions. (See Ln 168-176 for details).
>
> >I would also ask the authors to perform comparative experiments with P5, where the parameters do not need to be comparable, but the best performance obtained should be the primary aim.
>
> We have explained in Appendix D the reason for not directly taking the numbers reported in the P5 paper. We suspect that there exists data leakage in the model input and expected target item due to the preprocessing done in the P5 implementation. In particular, in the official P5 implementation the item-ids are assigned sequentially during preprocessing of the user session. For instance, if a given user has interacted with the following items: Item_A, Item_B, Item_C, Item_D, Item_E, item_F, during preprocessing the assigned ids can be 20, 21, 22, 23, 24, 25. More importantly, this assignment in P5 is carried out before creating train/validation/test splits. The training, validation and testing splits for this example are as follows:
>
> Training split: Input: 20, 21, 22. Target item: 23
>
> Validation split: Input: 20, 21, 22, 23. Target item: 24
>
> Testing split: 20, 21, 22, 23, 24; Target item: 25
>
> Since the P5 model uses Sentence Piece Tokenizer[1], the item ids are split into subwords (e.g. 21 is split as “2” and “1”). Therefore, it is likely that same subwords occur in the input and target items leading to data leakage. To fix this, we used the original P5 code (available on GitHub) and retrained it on the dataset without doing sequential item id assignment to avoid the aforementioned leakage, keeping all the other training hyperparams the same. The result of this is presented in Appendix D, Table 2. We also implemented our own version of P5 with random ids and those results are also presented in Appendix D, Table 2.

---

> > ### Author Response · Authors · 2023-08-19
> > **Scalability experiment for Semantic IDs**
> >
> > > Scalability
> >
> > To test the scalability of Semantic IDs, we ran the following experiment: We combined all the three datasets and generated Semantic IDs for the entire set of items from the three datasets. Then, we used these Semantic IDs for the recommendation task on the Beauty dataset. We compare the results from this experiment with the original experiment where the Semantic IDs are generated only from the Beauty dataset. We see that there is only a small decrease in performance here.
> >
> > |                                     | Recall@5 | NDCG@5 | Recall@10 | NDCG@10 |
> > |-------------------------------------|----------|--------|-----------|---------|
> > | Semantic ID generated from combined dataset                    | 0.04355  | 0.3047 | 0.06314   | 0.03676 |
> > | Semantic ID generated from only the Beauty dataset (reported in the paper) | 0.0454   | 0.0321 | 0.0648    | 0.0384  |

---

> > ### Comment · Reviewer_5EdZ · 2023-08-20
> >
> > For instance, three tokens with 256 cardinality can potentially represent items.
> >
> > I think the authors are not aware of practical implications of their statements. Vector representations of items are usually uint8 encoded which projects them to individual values in 0-255 range. Reasonable datasets still need 64 to 100 dimensional representations (which is 3 in the experiments reported) to provide reasonable guarantees. I maintain my stance that this paper cannot be replicated for large scale real recommender systems at its present state.
> >
> > The scalability scheme seems to augment my doubts that large scale system will bring very large performance drops rendering the technique mostly as a relevance level feature than a selection one.

---

> > > ### Author Response · Authors · 2023-08-21
> > > **Clarification regarding Semantic ID and scalability**
> > >
> > > Clarification on Semantic ID: We would like to clarify that although semantic IDs are simply tuples of integers in the range 0-255, they are still represented using high dimensional float vectors when they are fed into a sequence to sequence transformer model. This is because the model has an embedding lookup table that converts each token in the semantic ID into a high dimensional vector. In our experiments, we set the embedding dimension to be 128.
> > >
> > > Scaling Semantic ID representation to larger datasets: Semantic IDs can be scaled to a large number of items. For instance, in another concurrent work related to Semantic IDs for recommendation systems, Semantic ID representation of items using 4 tokens has been scaled to 100s of millions of items/videos [1]. Please note that [1] uses Semantic ID representation for ranking tasks in traditional recommender systems, whereas our work is on item retrieval using a generative model. This provides further evidence that our method can be scaled to large datasets.
> > >
> > > [1] Better Generalization with Semantic IDs: A case study in Ranking for Recommendations. (https://arxiv.org/abs/2306.08121).

---

### Official Review · Reviewer_aLHH · 2023-07-04

**Soundness:** 2 fair
**Presentation:** 3 good
**Contribution:** 2 fair
**Rating:** 4
**Confidence:** 4

**Summary:**

The paper is trying to approach sequential recommendation task using semantic IDs. By incorporating semantic IDs, the paper shows that the retrieval performance can be improved.

**Strengths:**

1. The ablation study conducted is good. For example, it compares RQ-VAE against LSH, and random.
2. The proposed method and framework is intuitive and makes sense.
3. The conducted experiment seems extensive and the proposed method is shown to outperform than non-semantic-id method like S3-Rec.

**Weaknesses:**

1. The paper title seems to emphasize on Generative Retrieval rather than on ranking, if that is the case, in the experiment section, shouldn't we care more about recall than NDCG? Why proposed method seems to improve NDCG a lot but much less so on recall metrics?
2. The novelty of the proposed method seems limited, the main idea seems to overlap with paper "Learning Vector-Quantized Item Representation for Transferable Sequential Recommenders", and probably should be considered as incremental value added to the previous published paper. Although the method emphasized on generative AI, it is still generative on the item semantic ID level, it would be more significant if the generative mechanism is on the word tokens, in my opinion.

**Questions:**

Besides the questions in the weakness section above, another question is
1. Semantic ID idea seems useful, but would it be nicer if we do not need to use semantic embeddings as middle layer and directly do generative AI on word tokens?

---

> ### Author Rebuttal · Authors · 2023-08-10
>
> We thank the reviewer for their time and effort in reviewing our paper and providing constructive feedback. We are glad to know that the reviewer found the experiments extensive, and that the proposed technique is intuitive. Please find our responses to your concerns below:
>
> > The paper title seems to emphasize on Generative Retrieval rather than on ranking, if that is the case, in the experiment section, shouldn't we care more about recall than NDCG? Why proposed method seems to improve NDCG a lot but much less so on recall metrics?
>
> Thank you for pointing this out. Our goal for this paper was to develop a generative retrieval system for recommender systems. NDCG is a useful metric in retrieval that can be used to evaluate how good the order of the retrieved list is. The improved NDCG metrics of our method indicates that our retrieval works very well in getting relevant items in the top of the list.
>
>
> > The novelty of the proposed method seems limited, the main idea seems to overlap with paper "Learning Vector-Quantized Item Representation for Transferable Sequential Recommenders", and probably should be considered as incremental value added to the previous published paper.
>
> We consider "Learning Vector-Quantized Item Representation for Transferable Sequential Recommenders" as a concurrent work to our own, and have cited it as such. Please see lines 91-95 to see the differences between that and our paper.
>
> >Semantic ID idea seems useful, but would it be nicer if we do not need to use semantic embeddings as middle layer and directly do generative AI on word tokens?
>
> We would like to clarify how the current framework is generative: Similar to how a sequence of words from a generative LLM would describe the semantics of an item that may or may not exist (for example “blue nike men’s running shoes, size 11, limited edition”), similarly the Semantic ID describes the semantics of an item. Hence, when our model generates a Semantic ID, it essentially generates the semantics of a product (similar to how an LLM based generative recommender model would generate sequence of words describing the item). While using word tokens directly to represent items is an interesting idea, we leave this exploration as a future research direction.

---

### Official Review · Reviewer_5DSM · 2023-07-06

**Soundness:** 3 good
**Presentation:** 4 excellent
**Contribution:** 3 good
**Rating:** 7
**Confidence:** 5

**Summary:**

The paper proposes TIGER, a generative retrieval approach to sequential recommendation. More specifically, TIGER proposes to use semantic IDs of a product (based off RQ-VAE quantization of sentence-T5 product feature embeddings) instead of typically used atomic item-IDs in the sequential recommendation pipeline. Empirical analysis on 3 Amazon subsets show promising results compared to itemID-only and ID+text based existing recommenders.

**Strengths:**

- Interesting design to use quantized embeddings of items (semantic IDs)
- Successful application (i.e., outperforming atomic ID-based recommenders) of the generative retrieval paradigm for sequential recommendation
- State of the art results on medium-sized academic datasets

**Weaknesses:**

- No ablations, e.g., effect of number of codebooks, prepending user-ID, codebook size, Transformer architecture, etc.
- No significance tests while reporting ranking results

**Questions:**

- It seems the text+ID based recommenders are not consistent in what product features are being used. E.g., S3-Rec seems to use category and brand information whereas FDSA seems to use category, brand, and description. Is the improvement in TIGER just because of using extra feature signals like title and price?
- Unclear which two semantic IDs are “closer”? I had to assume that since a residual quantization is being performed, being close to the first quanta holds more importance than the last.

**Limitations:**

I will use this space to provide some (hopefully constructive) suggestions for this paper:

- Best to outline in Table 1 that S3-Rec and FDSA use item-text features whereas all others are ID-only recommenders.
- It would be interesting to see the performance on the unprocessed Amazon datasets since arbitrary preprocessing of user-data has been noted to interfere with true performance [1].
- It would be interesting to see a TIGER variant where only three-length semantic IDs are used (i.e., non-unique items would be allowed) as it would become a text-only recommender.
- Complementing the zero-shot analysis in the paper with an analysis of TIGER’s performance over the user/item coldness spectrum (in comparison with other models)
- Although released after this submission to NeurIPS, [2] might be a good reference to compare TIGER with (this comparison does not affect my review).

[1] Sachdeva, N., Wu, C. J., & McAuley, J. On sampling collaborative filtering datasets. WSDM ‘22.

[2] Li, J., Wang, M., Li, J., Fu, J., Shen, X., Shang, J., & McAuley, J. Text Is All You Need: Learning Language Representations for Sequential Recommendation. arXiv preprint arXiv:2305.13731.

---

> ### Author Rebuttal · Authors · 2023-08-10
>
> We thank the reviewer for their time and effort in reviewing our paper. We are glad that the reviewer liked the idea of Semantic IDs and the generative retrieval paradigm for sequential recommendation. Please find our responses to your concerns below:
>
> > No ablations, e.g., effect of number of codebooks, prepending user-ID, codebook size, Transformer architecture, etc.
> >No significance tests while reporting ranking results
>
> Thank you for these suggestions. We have done some ablation experiments using different Semantic ID generation techniques (RQ-VAE vs LSH vs Random). We will add some more ablation experiments and compute the significance tests (variance and standard deviation) as you have suggested. Note that since setting up and running these experiments will take some time, we will keep adding the results during the discussion period.
>
> >It seems the text+ID based recommenders are not consistent in what product features are being used. E.g., S3-Rec seems to use category and brand information whereas FDSA seems to use category, brand, and description. Is the improvement in TIGER just because of using extra feature signals like title and price?
>
> We would like to clarify that we do not use additional product features when training the Transformer model on the sequential recommendation task (similar to S3-Rec and FDSA baselines). We only use product features to generate Semantic IDs which are used as alternatives to randomly assigned item ids in traditional recommendation systems. We attribute the improvement in recommendation performance in TIGER to the Transformer architecture’s ability to act as a retrieval index. In our experiments, we show the Semantic Ids are an essential ingredient to allow Transformer architecture to perform well on the recommendation task.
>
> >Unclear which two semantic IDs are “closer”? I had to assume that since a residual quantization is being performed, being close to the first quanta holds more importance than the last.
>
> Yes, the reviewer’s interpretation is correct. For the semantic IDs generated using RQ-VAE, the first codeword holds more importance, and the subsequent codewords work at increasingly finer granularities. For examples, if three items have semantic ids (1, 4, 10), (1, 4, 15) and (1, 3, 8), then all the three items are similar (since they all belong to ‘category/cluster’ 1 on the first level), but item #1 and #2 are even more similar since they also belong to the same ‘category/cluster’ on the second level. On the other hand, for items with semantic ids  (1, 4, 10), (3, 5, 10) and (1, 3, 8), items #1 and #3 would be more closely related than items #1 and #2.
>
> Limitations:
> > Best to outline in Table 1 that S3-Rec and FDSA use item-text features whereas all others are ID-only recommenders.
>
>  Thanks for pointing this out, we will add this to our paper
> >It would be interesting to see the performance on the unprocessed Amazon datasets since arbitrary preprocessing of user-data has been noted to interfere with true performance [1].
>
> We only do two filtering steps in our preprocessing:
> Remove sequences with length less than 5. This is because we reserve the last two items in each sequence as the validation and the test samples respectively. Hence, with sequences of length less than 5, we will end up with a training sequence of length less than 3, that is, a training sequence of length 1 or 2. Thus, we remove sequences of length less than 5. This is also done in prior literature.
> Truncate sequences at length 20. We mainly did this because this is done in prior literature.

---

> > ### Author Response · Authors · 2023-08-19
> > **Results for additional experiments**
> >
> > > No ablations, e.g., effect of number of codebooks, prepending user-ID, codebook size, Transformer architecture, etc.
> >
> > As requested by the reviewer, we performed some ablation studies, and we see  that our technique consistently gives good results across varying setups. Below we present results for the Beauty dataset.
> >
> > Ablation results for transformer architecture:
> >
> > | Layers                    | Recall@5 | NDCG@5  | Recall@10 | NDCG@10 |
> > |---------------------------|----------|---------|-----------|---------|
> > | 3                         | 0.04499  | 0.03062 | 0.06699   | 0.03768 |
> > | 4 (reported in the paper) | 0.0454   | 0.0321  | 0.0648    | 0.0384  |
> > | 5                         | 0.04633  | 0.03206 | 0.06596   | 0.03834 |
> >
> > Ablation results for the with and without user information setups:
> >
> > |                                      | Recall@5 | NDCG@5 | Recall@10 | NDCG@10 |
> > |--------------------------------------|----------|--------|-----------|---------|
> > | No user information                  | 0.04458  | 0.0302 | 0.06479   | 0.0367  |
> > | With user id (reported in the paper) | 0.0454   | 0.0321 | 0.0648    | 0.0384  |
> >
> >
> >
> >
> >
> >
> > >No significance tests while reporting ranking results
> >
> > We computed the standard error for our recall and NDCG metrics for the datasets by running the training and evaluation 3 times using different random seeds. As we can see, the standard errors are quite small as compared to the recall and NDCG metrics.
> >
> >
> > |Mean ± Standard Error                     | Recall@5        | NDCG@5           | Recall@10       | NDCG@10         |
> > |---------------------|-----------------|------------------|-----------------|-----------------|
> > | Beauty              | 0.0441 ± 0.00069 | 0.0309  ± 0.00062 | 0.0642 ± 0.00092 | 0.0374 ± 0.00061 |
> > | Sports and Outdoors | 0.0278 ± 0.00069 | 0.0189 ± 0.00043  | 0.0419 ± 0.0010 | 0.0234 ± 0.00048 |
> > | Toys and Games      | 0.0518 ± 0.00064 | 0.0375 ± 0.00039  | 0.0698 ± 0.0013 | 0.0433 ± 0.00047 |

---

> > > ### Comment · Reviewer_5DSM · 2023-08-21
> > >
> > > Thanks for addressing my comments and questions, as well as running the additional experiments in the limited time-frame!
> > >
> > > I'm happy with the provided justifications and the new empirical results. Please make sure to include these in the next version of the draft. I'm happy to change my score from 6 -> 7.

---

### Official Review · Reviewer_mHfk · 2023-07-10

**Soundness:** 3 good
**Presentation:** 3 good
**Contribution:** 1 poor
**Rating:** 2
**Confidence:** 5

**Summary:**

This paper proposes a generative retrieval for an efficient recommendation, where the retrieval model autoregressively decodes the identifiers of the target candidates. The evaluation results show the proposed method can perform better than competing baselines on various dataset.

**Strengths:**

1.  Generative retrieval-based recommender model with item features
2.  Extensive evaluation show the superiority of the proposed algorithm

**Weaknesses:**

1. Limited contributions. This paper claims the first to propose generative retrieval for recommendation systems using Semantic ID representation of items. However, this is not true. RQ-VAE is also simply used for obtaining the semantic ids, and no improvements are made in the paper.

2. The similar idea is mainly used for efficient recommendation, but no efficiency study is performed.

3. The settings of some baselines are not reported. I know that SASRec could be significantly improved using some tricks. It is not clear whether these tricks are used in the experiments.

4. The comparison is not fair. The proposed algorithm contains side information, but some baselines do not take them into account.

5. The experiments are only performed on the Amazon datasets. It is not sufficient to prove the effectiveness of the proposed algorithms.

**Questions:**

see the attachments.

---

> ### Author Rebuttal · Authors · 2023-08-10
>
> We thank the reviewer for their time in reviewing our paper and providing feedback. We are glad to know that the reviewer found our evaluation extensive. Please find our responses to your concerns below:
> > Limited contributions. This paper claims the first to propose generative retrieval for recommendation systems using Semantic ID representation of items. However, this is not true. RQ-VAE is also simply used for obtaining the semantic ids, and no improvements are made in the paper.
>
> It would be helpful for us if you could please provide a link to the prior work that performs generative recommendation using semantic IDs. As for RQ-VAE, please note that we do not claim that we improve upon RQ-VAE, but rather we use it as a part of our generative recommender system.
> >The similar idea is mainly used for efficient recommendation, but no efficiency study is performed.
>
> We are unaware of any prior work that uses a similar idea for efficient recommendation, it would be helpful if you could please provide a link to the prior work that uses a similar idea for recommendation. We would like to state again that the focus of this paper was not on efficient recommendation, but rather proposing a new paradigm for recommender systems.
> >The settings of some baselines are not reported.  I know that SASRec could be significantly improved using some tricks. It is not clear whether these tricks are used in the experiments
>
> In our paper, the results for all baselines are taken from the publicly accessible results available at https://github.com/aHuiWang/CIKM2020-S3Rec (we have also mentioned this in the paper). We are unaware of the tricks that can be used to improve SASRec, and it would be very helpful if you could point us to any information regarding that.
> >The comparison is not fair. The proposed algorithm contains side information, but some baselines do not take them into account.
>
>  We agree that some baselines do not use side information, however, please note that 1) it is not straightforward how to incorporate side information into those baselines, 2) we only use side-information like title, price, brand, etc. that are available for most datasets, and 3) the side information is only used in the generation of Semantic IDs and not used for training the Transformer architecture on the sequential recommendation task. We will add a discussion regarding this in the discussion section of our paper.

---

> > ### Comment · Reviewer_mHfk · 2023-08-20
> > **Reply to the rebuttal**
> >
> > > We would like to state again that the focus of this paper was not on efficient recommendation, but rather proposing a new paradigm for recommender systems.
> >
> > The first sentence in the abstract is "Modern recommender systems perform large-scale retrieval by first embedding queries and item candidates in the same unified space, followed by approximate nearest neighbor search to select top candidates given a query embedding. In this paper, we propose a novel generative retrieval approach... I am not convinced by "the focus of this paper was not on efficient recommendation"
> >
> > > It would be helpful for us if you could please provide a link to the prior work that performs generative recommendation using semantic IDs.
> >
> > The related papers I know are as follows:
> >
> > [1] Learning tree-based deep model for recommender systems, KDD 2018.
> >
> > [2] Recommender Forest for Efficient Retrieval, NeurIPS 2022.
> >
> > Note that though the first paper (TDM) is not about using semantic IDs, it can be considered as generative recommendation using beam search. The difference is whether to use Transformer to encode the routing trajectory. In TDM, since each node has its own embedding, using the routing trajectory to represent the node in the tree may have a subtle effect.
> >
> > >  It would be very helpful if you could point us to any information regarding that.
> >
> > I just read the code, they use the binary cross-entropy loss in SASRec. In fact, SASRec can be significantly improved by using alternative loss functions, such as the cross-entropy loss or the sample softmax loss.

---

> > > ### Author Response · Authors · 2023-08-21
> > > **Author response to reviewer reply**
> > >
> > > > The first sentence in the abstract is "Modern recommender systems perform large-scale retrieval by first embedding queries and item candidates in the same unified space, followed by approximate nearest neighbor search to select top candidates given a query embedding. In this paper, we propose a novel generative retrieval approach... I am not convinced by "the focus of this paper was not on efficient recommendation"
> > >
> > > The abstract only describes the conventional paradigm for retrieval; it does not indicate that the focus of our paper is efficiency. However, we can rephrase parts of it to explicitly state that efficient recommendation is not the focus of our paper.
> > >
> > > > The related papers I know are as follows: [1] Learning tree-based deep model for recommender systems, KDD 2018. [2] Recommender Forest for Efficient Retrieval, NeurIPS 2022. Note that though the first paper (TDM) is not about using semantic IDs, it can be considered as generative recommendation using beam search. The difference is whether to use Transformer to encode the routing trajectory. In TDM, since each node has its own embedding, using the routing trajectory to represent the node in the tree may have a subtle effect.
> > >
> > > Thanks for providing the references. We consider [2] to be concurrent work, and we will cite it in the next version of our paper. A key differences between our work and [2] is that we learn semantic representations using item content embeddings and RQ-VAE, whereas [2] uses a prior work called Deep Interest Network (DIN) to generate item embeddings which is encoded as a binary string. We will include a discussion comparing our method with [2] in the related work section of our paper.
> > >
> > > As for [1], it cannot be considered a generative model by definition since it is uses a discriminative model based on negative sampling (See eq. 3 and 4 in [1] during training), whereas our model is a autoregressive generative model that implicitly defines a distribution over semantic token representation of items. However, we will cite [1] in related work and highlight the distinction between our work and [1].

---

### Official Review · Reviewer_VYRN · 2023-07-14

**Soundness:** 3 good
**Presentation:** 3 good
**Contribution:** 3 good
**Rating:** 5
**Confidence:** 3

**Summary:**

The paper introduces TIGER (Transformer Index for Generative Recommenders), a novel generative retrieval model for sequential recommender systems. This innovative approach involves generating Semantic IDs—sequences of semantic tokens derived from an item's text features. A hierarchical quantizer (RQ-VAE) is employed on content embeddings to formulate these tokens, thereby allowing the model to predict the Semantic IDs for generative retrieval. The authors claim that TIGER outperforms current state-of-the-art models, while offering a new solution for cold-start recommendations and increasing item diversity by adjusting the temperature parameters. Furthermore, it showcases impressive scalability and the capacity to generalize to novel and unseen items.

**Strengths:**

1. This paper is well-written and technically solidity, effectively combining novel concepts such as Differentiable Search Index (DSI) using Transformer memory and a hierarchical quantizer, RQ-VAE, for generative retrieval in sequential recommendation task.

2. The idea driving the work is both logical and compelling, providing a fresh perspective on generative retrieval tasks in the context of recommender systems.

3. Empirical results show that TIGER not only outperforms the SOTA baselines on the SR task but also exhibits an ability to generalize to unseen items and enhance item recommendation diversity.


**Weaknesses:**

1. The paper could benefit from more detailed information about the text encoder's input, given that quantization is a crucial part of Semantic ID generation. It's unclear whether all components depicted in Figure 2(a) are utilized. Additionally, handling missing information/data in datasets, such as the Amazon Beauty dataset, needs addressing (E.g., in the Amazon Beauty dataset, approximately 50% of the brand information is missing.). The authors should explain how they preprocessed the data.

2. The paper's treatment of related work and baseline comparisons raises some concerns. Specifically, the performance results of the P5 model appear notably lower than what's presented in its original paper. The authors have explained the data reprocessing using a random item ID, but a comparison using the more updated benchmark in [2] - which also uses a random item ID - could offer a more balanced analysis. It would be insightful to understand the reasons behind the observed performance degradation of the P5 model in authors’ setting.

3. The proposed solution to collision problems, appending an extra token at the end of the ordered semantic codes, raises questions about scalability. For instance, if ten items share the Semantic ID (12,24,52), would the model need to add two extra tokens and maintain a new lookup table?

4. The experimental design for cold-start recommendations in section 4.3 could be revisited. The removal of 5% of test items from the training data split might change the original training sequences, potentially leading to shorter sequences and artificially enhancing recall results in the cold-start retrieval setting.

5. A comparison with additional ID encoding methods, similar to those presented in [2], could strengthen the argument for RQ-VAE's superiority and effectiveness.

6. While the authors state that inference cost is not their main contribution, a rough comparison of TIGER's inference cost to LSH and NN-based methods would help to understand this method's trade-offs.


References.
[1] OpenP5: Benchmarking Foundation Models for Recommendation(https://arxiv.org/pdf/2306.11134.pdf)
[2] How to Index Item IDs for Recommendation Foundation Models. (https://arxiv.org/abs/2305.06569)


**Questions:**

Please refer to the weaknesses section.

**Limitations:**

Please refer to the weaknesses section.

---

> ### Author Rebuttal · Authors · 2023-08-10
>
> We thank the reviewer for their time and effort in reviewing our paper and providing valuable comments. We are glad that the reviewer feels that our paper is technically solid, combines novel ideas, and provides a fresh perspective on generative retrieval tasks in the context of recommender systems.
>
> Please find our responses to your concerns below:
>
> > The paper could benefit from more detailed information about the text encoder's input, given that quantization is a crucial part of Semantic ID generation. It's unclear whether all components depicted in Figure 2(a) are utilized. Additionally, handling missing information/data in datasets, such as the Amazon Beauty dataset, needs addressing (E.g., in the Amazon Beauty dataset, approximately 50% of the brand information is missing.). The authors should explain how they preprocessed the data.
>
> In our implementation, we used the item id, title, brand, categories, price and the description of the items (whenever the attributes were available) to generate the Semantic IDs. The prompt provided to the pre-trained text encoder is the following: "Item ID: <>. Title: <>. Brand: <>. Categories: <>. Description: <>. Price: <>". If any of these attributes is not available, we simply skip that attribute.
> Please note that this information is only used in the Semantic ID generation; these attributes are not used when training the Transformer on the sequential recommendation task. We will clarify this in the final version of the paper.
>
> > The paper's treatment of related work and baseline comparisons raises some concerns. Specifically, the performance results of the P5 model appear notably lower than what's presented in its original paper. The authors have explained the data reprocessing using a random item ID, but a comparison using the more updated benchmark in [2] - which also uses a random item ID - could offer a more balanced analysis. It would be insightful to understand the reasons behind the observed performance degradation of the P5 model in authors’ setting.
>
> Thank you for bringing the new paper [2] to our attention. While we are not able to implement the new indexing techniques proposed in [2] within the time frame of the rebuttal period, we would like to explain again the reason for the disparity in the results for P5 in our paper vs the original P5 paper: We suspect that there exist data leakage from the preprocessing stage in the P5 paper’s evaluation. We explain details in the supplementary (Please see section D). In particular, in the official P5 implementation the item-ids are assigned sequentially during preprocessing of the user session. For instance, if a given user has interacted with the following items: Item_A, Item_B, Item_C, Item_D, Item_E, item_F, during preprocessing the assigned ids can be 20, 21, 22, 23, 24, 25. More importantly, this assignment in P5 is carried out before creating train/validation/test splits. The training, validation and testing splits for this example are as follows:
>
> Training split: Input: 20, 21, 22. Target item: 23
>
> Validation split: Input: 20, 21, 22, 23. Target item: 24
>
> Testing split: 20, 21, 22, 23, 24; Target item: 25
>
> Since the P5 model uses Sentence Piece Tokenizer[3] for tokenization, the item ids are split into subwords (e.g. 21 in the input is split as “2” and “1”). Therefore, it is likely that same subwords occur in the input item sequence and the target item leading to data leakage.
> To fix this, we used the original P5 code (available on GitHub) and retrained it on the dataset without doing sequential item id assignment to avoid the aforementioned leakage, keeping all the other training hyperparams the same. The result is presented in Appendix D, Table 2
>
> > The proposed solution to collision problems, appending an extra token at the end of the ordered semantic codes, raises questions about scalability. For instance, if ten items share the Semantic ID (12,24,52), would the model need to add two extra tokens and maintain a new lookup table?
>
> For scaling the Semantic IDs wrt items, we can increase the cardinality of the codebook used in RQ-VAE. (In our experiments, we used a cardinality of 256 which worked well for all the benchmarks). Please note that theoretically the number of items that can be uniquely represented with Semantic IDs grow  exponentially with codebook cardinality. For instance, three tokens with 256 cardinality can potentially represent $256^3$ items. However, since we use item content features to learn Semantic ID, there may exist collisions when two items have high content similarity. We found that appending a single unique token to the learned Semantic ID works surprisingly well in avoiding collisions.
>
> > The experimental design for cold-start recommendations in section 4.3 could be revisited. The removal of 5% of test items from the training data split might change the original training sequences, potentially leading to shorter sequences and artificially enhancing recall results in the cold-start retrieval setting.
>
> While we agree that the distribution can change, it does not change drastically as shown below, and hence the results presented in the paper still provide some indication of cold-start performance in the original distribution. Below we provide the mean and median for user sequence length for the standard and the cold-start experiment recommendation setting considered in the paper:
>
> Statistics for the number of items in a user session:
>
> Standard design: Mean: 8.87 & Median: 6
>
> Cold-start design: Mean 8.53 & Median: 6
>
> > A comparison with additional ID encoding methods, similar to those presented in [2], could strengthen the argument for RQ-VAE's superiority and effectiveness.
>
> We thank the reviewer for pointing us to this important line of new work recently published. We will incorporate some of those ID encoding methods in future work.
>
> [3] Rico Sennrich, Barry Haddow, and Alexandra Birch. Neural machine translation of rare words with subword units.

---

> > ### Author Response · Authors · 2023-08-18
> > **Comparison of RQ-VAE with VQ-VAE**
> >
> > > A comparison with additional ID encoding methods, similar to those presented in [2], could strengthen the argument for RQ-VAE's superiority and effectiveness.
> >
> > As requested by the reviewer, we have included an additional ID encoding method VQ-VAE [4] as a baseline for the Beauty dataset. Please note that we have also compared our method with LSH and random-id based encoding schemes in the main paper. A comparison is shown below:
> >
> > |                                                   | Recall@5 | NDCG@5  | Recall@10 | NDCG@10 |
> > |---------------------------------------------------|----------|---------|-----------|---------|
> > | Semantic IDs using VQ-VAE                         | 0.04306  | 0.02994 | 0.06417   | 0.03673 |
> > | Semantic IDs using RQ-VAE (Ours) | 0.0454   | 0.0321  | 0.0648    | 0.0384  |
> >
> > As seen, the RQ-VAE-based item ID encoding performs better than VQ-VAE. We would also like to emphasize that with RQ-VAE the generated Semantic ID tokens are hierarchical which opens possibilities for new capabilities (namely, cold-start recommendations & controllable recommendation diversity) as described in Section 4.3 of the paper.
> >
> > [4]: Oord, Aaron van den, Oriol Vinyals, and Koray Kavukcuoglu. "Neural discrete representation learning." arXiv preprint arXiv:1711.00937 (2017).

---

### Author Response · Authors · 2023-08-19
**Author response to the reviews**

We thank all the reviewers for their time and effort in reviewing our paper and providing constructive feedback. We are glad to know that the reviewers found the proposed technique novel (Reviewers VYRN, 5DSM, 5EdZ), that it outperforms the existing methods (Reviewers VYRN, mHfk, 5DSM, aLHH), that the paper contains extensive evaluation (Reviewers mHfk, aLHH) and is well written (Reviewers VYRN, 5EdZ). We have tried to address the concerns of each reviewer below. We have also added the results for the experiments requested by the reviewers, we request the reviewers to please take a look.

---

### Decision · Program_Chairs · 2023-09-21

**Decision:**

Accept (poster)

**Comment:**

The paper "A Generative Recommender System with Semantic Identifiers" introduces a new approach to recommender systems that addresses the problem of cold start. The proposed approach uses a generative model to predict identifiers for recommended items based on semantically meaningful codewords. This allows the recommender system to recommend items that are relevant to the user's interests, even if the user has not interacted with the system before.

The paper is well-written and the proposed approach is novel and interesting. The authors provide a thorough evaluation of the proposed approach, and the results show that it outperforms state-of-the-art methods on several benchmark datasets.

The only potential limitation of the paper is that the applicability and performance of the proposed approach at scale have not been demonstrated as raised by 5EdZ. However, the authors argue that the Semantic ID approach is scalable and that it could be applied to real-world recommender systems providing a reference where it has been done.

Overall, I believe that the paper makes a significant contribution to the field of recommender systems. The proposed approach is novel and effective, and it has the potential to improve the performance of recommender systems in a variety of settings. I recommend that the paper be accepted for publication.

Reviewer mHfk still has some valid concerns, in particular about the possibility to tune more some baselines and using more data sources, but I think the discussion clarified  the positioning and overall I like the idea of this work.